# Novel Hybrid Electrode Coatings Based on Conjugated Polyacid Ternary Nanocomposites for Supercapacitor Applications

**DOI:** 10.3390/molecules28135093

**Published:** 2023-06-29

**Authors:** Sveta Ozkan, Lyudmila Tkachenko, Valeriy Petrov, Oleg Efimov, Galina Karpacheva

**Affiliations:** 1A.V. Topchiev Institute of Petrochemical Synthesis, Russian Academy of Sciences, 29 Leninsky Prospect, Moscow 119991, Russia; ozkan@ips.ac.ru (S.O.); petrov@ips.ac.ru (V.P.); 2Federal Research Center of Problems of Chemical Physics and Medicinal Chemistry, Russian Academy of Sciences, 1 Academician Semenov Avenue, Chernogolovka 142432, Russia; bineva@icp.ac.ru (L.T.); efimor@yandex.ru (O.E.)

**Keywords:** polydiphenylamine-2-carboxylic acid, binary and ternary nanocomposites, activated IR-pyrolyzed polyacrylonitrile, single-walled carbon nanotubes, hybrid electrode coatings, organic electrolyte

## Abstract

Electrochemical behavior of novel electrode materials based on polydiphenylamine-2-carboxylic acid (PDPAC) binary and ternary nanocomposite coatings was studied for the first time. Nanocomposite materials were obtained in acidic or alkaline media using oxidative polymerization of diphenylamine-2-carboxylic acid (DPAC) in the presence of activated IR-pyrolyzed polyacrylonitrile (IR-PAN-a) only or IR-PAN-a and single-walled carbon nanotubes (SWCNT). Hybrid electrodes are electroactive layers of stable suspensions of IR-PAN-a/PDPAC and IR-PAN-a/SWCNT/PDPAC nanocomposites in formic acid (FA) formed on the flexible strips of anodized graphite foil (AGF). Specific capacitances of electrodes depend on the method for the production of electroactive coatings. Electrodes specific surface capacitances C_s_ reach 0.129 and 0.161 F∙cm^−2^ for AGF/IR-PAN-a/PDPAC_ac_ and AGF/IR-PAN-a/SWCNT/PDPAC_ac_, while for AGF/IR-PAN-a/PDPAC_alk_ and AGF/IR-PAN-a/SWCNT/PDPAC_alk_ C_s_ amount to 0.135 and 0.151 F∙cm^−2^. Specific weight capacitances C_w_ of electrodes with ternary coatings reach 394, 283, 180 F∙g^−1^ (AGF/IR-PAN-a/SWCNT/PDPAC_ac_) and 361, 239, 142 F∙g^−1^ (AGF/IR-PAN-a/SWCNT/PDPAC_alk_) at 0.5, 1.5, 3.0 mA·cm^−2^ in an aprotic electrolyte. Such hybrid electrodes with electroactive nanocomposite coatings are promising as a cathode material for SCs.

## 1. Introduction

With the development of alternative resource-saving energy, it is of great importance to solve the problems related to the creation of both energy storage devices and new ways to store energy [1,2,3]. The popularity of wearable electronic devices cause interest in the development of more efficient energy storage systems [4,5,6,7,8]. The supercapacitors (SCs) of high power density, long service life and comparatively high maintenance attract special attention among such systems [9,10,11,12]. While batteries provide better energy density for storage, SCs ensure faster charge and discharge [7,13,14]. Therefore, SCs are used when no large energy storage capacity is required, but powerful impulses are wantrequired, such as for starting electric cars.

The combination of advantages of batteries and SCs can be achievable in the hybrid SCs based on conductive polymers and carbon nanomaterials, where energy is stored at the electrode/electrolyte interface using double-layer capacitance and Faraday pseudocapacitance [15,16,17,18,19,20,21,22,23,24]. This type of a system with two energy storage mechanisms requires the development of highly efficient electrode materials that determine the capacitance, energy density, and specific power of SCs [25,26,27,28,29]. The use of electro-active conductive polymers (ECP) in the creating of electrode materials of hybrid SCs is an undoubted achievement of the last decade [30,31]. Quasi-reversible electrochemical charge-discharge processes in such polymers are carried out when they are doped with counter-ions due to the formation of delocalized π-electrons or holes and their transfer under the influence of an electric field along the system of conjugated double bonds of ECP. In terms of specific energy and specific power, the electrochemical SCs with ECP are in an intermediate position between the double-layer SCs and the lithium-ion batteries. Such polymer-carbon nanomaterials as electroactive coatings for hybrid electrodes are promising for the creation of SCs, rechargeable batteries, electrochemical current sources, solar panels, fuel cells, etc. [32,33,34,35,36,37,38,39,40,41].

Graphene-like materials and carbon nanotubes can be used as carbon materials for hybrid electrode coatings [34,38,42,43,44,45]. They easily form composites with redox active polymers to increase the pseudocapacitance for energy storage [46,47,48,49]. Along with carbon nanoparticles, porous carbon materials, in particular activated carbon materials, are considered promising [21,25,28,29]. The latter are derived from carbon precursors by physical or chemical activation.

The use of activated IR-pyrolyzed polyacrylonitrile (IR-PAN-a) offers exciting electrochemical properties of resulting composites by changing the structural characteristics of the carbon material. The carbon structure of IR-PAN-a contains nitrogen atoms that provide additional Faraday pseudocapacity [50,51]. The introduction of nitrogen heteroatoms into the structure of carbonaceous materials leads to an increase in the free space for electrolyte placement, providing high accessibility of electrolyte ions to the active surface [40,52].

In the last decade, researchers have focused on the production of hybrid electrode materials that include, along with a conductive polymer, a combination of different carbon nanomaterials. Only a few papers have shown that PANI-based nanocomposites containing graphene and CNT nanoparticles have improved SCs [53,54,55]. PANI is the most studied electro-active polymer. Its main advantages are simplicity of synthesis, ease of doping-dedoping processes, stability of properties. The simultaneous presence of two different carbon nanoparticles in the nanocomposite composition contributes to the formation of a three-dimensional structure where redox centers of PANI are more accessible. The addition of CNT to two-dimensional graphene nanosheets increases the surface area, but physical interactions between carbon nanomaterials are not enough to hold them together. PANI is considered as a kind of glue for attaching various types of carbon nanomaterials to each other [56]. Not only the performance of SCs is significantly increased, but also the resistance to charge transfer is reduced in such a structure.

In order to obtain both high capacitance and high charge-discharge currents, the combination of a high porous carbon substrate, carbon nanoparticles and electroactive polymer in electrode materials makes it possible to balance contributions of double layer charging and Faraday pseudocapacitance.

Attention should be drawn to the fact that the information available in the literature on the study of electrode materials based on ternary nanocomposites of PANI with two carbon components describes the results of electrochemical measurements, conducted in acid or alkaline aqueous electrolytes. We have not been able to find any references to studies of such nanocomposites in lithium organic electrolytes. The main disadvantages of aqueous electrolytes are low discharge voltage, narrow operating temperature range, high corrosive activity. On the other hand, lithium batteries use organic electrolytes, which have a wider range of operating potentials and operating temperatures, high corrosion resistance. It should be noted that the number of works on SCs with organic electrolytes is extremely limited (some of them were done by the authors of this article [57,58]). Moreover, we have not been able to find any research work on electrode materials based on ECP ternary nanocomposites with two carbon components in lithium organic electrolytes. Nevertheless, the transition to organic electrolytes gives the prospect of creating hybrid devices that combine the advantages of SCs and lithium batteries.

The present work is the first study of the electrochemical behavior of a cathode material based on a conjugated polyacid, IR-PAN-a and CNT in a lithium organic electrolyte. Such cathode materials in organic electrolytes are the most promising for the creation of hybrid SCs due to the possibility of increasing the SC voltage and achieving high values of energy density and charge-discharge current.

In this research paper, for electroactive coatings, hybrid ternary polymer-carbon-carbon nanocomposites based on polydiphenylamine-2-carboxylic acid (PDPAC), IR-PAN-a and SWCNT were prepared in two different ways for the first time. The choice of PDPAC as a polymer component is due to the possibility of coordination of carbon nanoparticles not only for amine but also for carboxylic groups. Furthermore, the presence of carboxylic groups makes it difficult to aggregate polymer chains (as in the case of PANI). Steric difficulties caused by carboxylic groups contribute to a looser structure that facilitates electrolyte penetration.

SWCNTs, which differ in high electrical conductivity and graphite-like external surface of the walls, are selected as one of the carbon components. Due to the π-π* stacking, adsorption of aromatic monomers is possible on such a surface, during oxidative polymerization of which a polymer shell grows, which prevents CNT aggregation. CNTs form an internal conductive framework that provides electronic transport in the composite.

Another carbon component of polymer-carbon-carbon nanocomposite is IR-PAN, leached to form a highly porous structure, adsorbing a part of the polymer phase, resulting in the general loosening of the nanocomposite.

In this work, the IR-PAN-a/SWCNT/PDPAC nanomaterials were synthesized in an acidic medium or in an alkaline medium via in situ oxidative polymerization of DPAC monomer in the presence both of SWCNT and IR-PAN-a as a highly porous N-doped carbon component. Electrochemical properties of the ternary IR-PAN-a/SWCNT/PDPAC nanocomposite electroactive coatings on a flexible strips of AGF with a developed porous surface in 1 M LiClO_4_ in propylene carbonate organic electrolyte were investigated. For comparison, under the same conditions, the binary composite coatings of IR-PAN-a/PDPAC were studied.

## 2. Results and Discussion

### 2.1. Synthesis and Characterization of IR-PAN-a/SWCNT/PDPAC Nanocomposites

PDPAC-based ternary nanocomposites were prepared via in situ oxidative polymerization of DPAC in the presence of IR-PAN-a and SWCNT in acidic and alkaline media. For comparison, binary composites of IR-PAN-a/PDPAC were obtained under the same conditions. Figure 1 shows a synthesis scheme of ternary nanomaterials of IR-PAN-a/SWCNT/PDPAC.

The formation of IR-PAN-a/SWCNT/PDPAC nanomaterials was confirmed by FTIR and Raman spectroscopy, XRD and field emission scanning electron microscopy (FE-SEM).

According to XRD analysis, as well as IR-PAN-a/PDPAC, IR-PAN-a/SWCNT/PDPAC nanocomposites are amorphous irrespective of the preparing method (Figure 2). Diffractograms of binary and ternary composites identify reflection peaks of IR-PAN-a in the range of scattering angles 2θ = 39°, 69° (Cr*K*_α_ radiation). These diffraction peaks correlate to Miller indices (002), (101). The carbon phase reflection peak from a single SWCNT plane is not identified.

Figure 3 shows the ATR FTIR spectra of the ternary nanocomposites obtained in an acidic medium (IR-PAN-a/SWCNT/PDPAC_ac_) and in an alkaline medium (IR-PAN-a/SWCNT/PDPAC_alk_). The main bands of IR-PAN-a/SWCNT/PDPAC and IR-PAN-a/PDPAC are the same [59,60]. The chemical structure of the polymer component mainly depends on the pH of the reaction medium for the nanocomposite synthesis (Figure 1). In the IR-PAN-a/PDPAC_ac_ composite, the absorption bands at 751, 785, and 892 cm^–1^ are due to the out-of-plane bending vibrations of the δ_C–H_ bonds of the 1,2-, 1,2,4-, and 1,4-substituted benzene rings, respectively. In the IR-PAN-a/PDPAC_alk_ composite, the absorption bands at 745 and 820 cm^–1^ correspond to out-of-plane bending vibrations of the δ_C–H_ bonds of the 1,2-disubstituted and 1,2,4-trisubstituted benzene rings. In the ternary IR-PAN-a/SWCNT/PDPAC nanocomposites, the shift in the absorption bands, corresponding to stretching vibrations of ν_C–C_ bonds in the aromatic rings indicate the π–π* interaction of PDPAC phenyl rings with the aromatic structures of IR-PAN-a and SWCNT (stacking effect). The charge transfer from the polymer chain to IR-PAN-a/SWCNT is manifested in the shift of skeletal oscillation frequencies of the polymer component.

Figure 4 shows the Raman spectra of the IR-PAN-a/PDPAC and composites prepared using two methods. As can be seen, in the Raman spectra of IR-PAN-a-based composites, there are two pronounced G and D bands. A G band at ~1596 cm^−1^ characterizes sp^2^ carbon atoms. A D band at ~1339 cm^−1^ corresponds to sp^3^ carbon atoms. The G band is a distinctive feature of graphite structures, whereas the D band is associated with disordered and defective structures [43]. The intensity ratio of these bands in the Raman spectrum of neat IR-PAN-a is I_D_/I_G_ = 0.91 [59]. The splitting of the G and D bands in the Raman spectra of the binary and ternary composites is associated with the presence of a polymer component. In the IR-PAN-a/SWCNT/PDPAC, the intensity ratio of the I_D_/I_G_ decreases to 0.04 regardless of the method of obtaining composites due to sp^2^ carbon atoms of SWCNT.

Figure 5 shows FE-SEM images of IR-PAN-a/SWCNT/PDPAC both as powders and as coatings on AGF. Good adhesion of composite films to the loosened surface of the AGF substrate makes it possible to create electroactive coatings on it by pouring stable dispersions of composites in FA [57,58].

According to FE-SEM data, in ternary nanocomposites (Figure 5a,c), the presence of carbon nanotubes contributes to the loosening of materials. Nanocomposites are permeated with SWCNT with a polymer film coating due to π-π* interaction of phenyl rings of PDPAC with SWCNT aromatic structures (stacking effect) [61]. The formation of a polymer coating on the surface of CNT provides interfacial charge transport. The electrical conductivity of ternary IR-PAN-a/SWCNT/PDPAC nanocomposites is higher than that of binary composites, which is associated with the presence of CNT. Regardless of the synthesis method, the electrical conductivity of IR-PAN-a/SWCNT/PDPAC reaches (4.8–7.2) × 10^−3^ S/cm (Table 1).

Thick (80 nm) interwoven bundles are visible in the nanocomposite coating which forms on AGF after pouring the suspension of IR-PAN-a/SWCNT/PDPAC_ac_ in FA subjected to ultrasonic treatment. These bundles are SWCNT with a polymer coating, with adhering particles of the IR-PAN-a/PDPAC_ac_ composite (Figure 5b). This results in a three-dimensional hierarchical porous structure of the IR-PAN-a/SWCNT/PDPAC_ac_ coating on AGF and an increase in the surface area available for electrolyte wetting.

In the nanocomposite coating of IR-PAN-a/SWCNT/PDPAC_alk_ on AGF (Figure 5d), polymer-coated SWCNT and IR-PAN-a are dispersed in the polymer matrix. Polymer-coated fragments of IR-PAN-a are engaged in the three-dimensional structure of the nanocomposite. On the surface of this film coating there are cavities, which leads to an increase in the surface of the composite contacting with the electrolyte.

### 2.2. Electrochemical Behavior of Nanocomposite Coatings in an Organic Electrolyte

Structural features of nanocomposite coatings are clearly reflected on CV when IR-PAN-a/SWCNT/PDPAC suspensions in FA are applied to the smooth surface of GC. Here, unlike in case of binary nanocomposites (IR-PAN-a/PDPAC and SWCNT/PDPAC), there is practically no transition of the active mass of ternary coatings into the electrolyte. Figure 6 shows CV and charge-discharge curves of the electrodes of GC/IR-PAN-a/SWCNT/PDPAC in 1 M LiClO_4_ in propylene carbonate at the potential scan rate of 20 mV·s^−1^ in the potential range from −0.5 V to 1.3 V.

The CV of GC/IR-PAN-a/SWCNT/PDPAC_ac_ electrode material from cycle 2 demonstrates an anodic peak at 1.0 V and a cathodic peak at 0.75 V, that refer to the formation of the PDPAC_ac_^2+^ dication (Figure 6a). Redox transitions with the formation of the PDPAC_ac_^+·^ radical cation are not identified on CV. In the region of 0.7 V, there is a sharp increase in current on the anodic branch. This is connected with the merging of two redox transitions into one due to the increase in electronic and ionic conductivity for doped PDPAC_ac_. The CV shape and peak position of the IR-PAN-a/SWCNT/PDPAC_ac_ composite coating on GC is identical to that of the PDPAC_ac_ polymer coating (1.02 V and 0.78 V) [57]. For IR-PAN-a/SWCNT/PDPAC_ac_ composite, Coulombic efficiency ŋ increases to 98% (Table 2), while for PDPAC_ac_ polymer coating, Coulombic efficiency ŋ is 84%. When cycling anode and cathode currents fall very slowly. Practically, the transfer of the electroactive mass into the electrolyte does not occur.

The CV of GC/IR-PAN-a/SWCNT/PDPAC_alk_ composite electrode shows two redox transitions in the polymer (Figure 6c). In the range of lower potentials from 0.25 to 0.75 V, the blurred anodic peak characterizes the transition of the polymer to the PDPAC_alk_/PDPAC^+∙^ radical cation state. Very broad cathodic peak in the range from −0.25 V to +0.25 V refers to the reduction of radical cation centers, some of which are located in the near electrode layer due to the dissolution of the coating. Very clear transition at 0.87 V associated with further oxidation of radical cations to PDPAC^+∙^/PDPAC^2+^ dication. A cathodic peak is observed at 0.78 V for the PDPAC^2+^/PDPAC^+∙^ redox transition. When cycling IR-PAN-a/SWCNT/PDPAC_alk_ composite coating, the anode and cathode currents decrease due to the transition of the electroactive mass into the electrolyte. For GC/IR-PAN-a/SWCNT/PDPAC_ac_ and GC/IR-PAN-a/SWCNT/PDPAC_alk_, the electrochemical capacitances of composite coatings calculated from CV at a potential scan rate of 20 mV·s^−1^ are 44 and 37 F∙g^−1^ with Coulombic efficiency ŋ = 98 and 93%, respectively (Table 2).

The comparison of CV of ternary nanocomposites obtained in different media in the simultaneous presence of IR-PAN-a (10 wt%) and SWCNT (10 wt%) (Figure 6a,c) shows that the capacitance of the IR-PAN-a/SWCNT/PDPAC_alk_ nanocomposite coating on GC is determined by a larger contribution of double-layer charge-discharge. This is expressed in a sharp increase in the area of CV in the range of potentials from −0.5 to +0.6 V. Whereas the capacitance of IR-PAN-a/SWCNT/PDPAC_ac_ is determined by a larger contribution of the Faraday pseudocapacitance. These features are also manifested in the shape of charge-discharge curves (Figure 6b,d).

For comparable coating weights on GC, for GC/IR-PAN-a/SWCNT/PDPAC_alk_, the capacitances calculated from the charge-discharge curves are 35 and 29 F∙g^−1^ at 0.1 and 0.5 mA∙cm^−2^ with a capacity loss of 8% in the first 10 cycles at a charge-discharge current of 0.1 mA∙cm^−2^. For the IR-PAN-a/SWCNT/PDPAC_ac_ composite coating, higher capacitance values of 40 and 38 F∙g^−1^ were obtained under the same conditions with a 5% loss of capacity in the first 10 cycles. This may be due to the higher conductivity of the composite in which the polymer is in a finished state (Table 1). At a charge-discharge current of 0.5 mA∙cm^–2^, the electrodes operate stably over 50 cycles. On AGF, the capacitance values of these coatings at 0.5 mA∙cm^−2^ increase by an order of magnitude and amount to 394 F∙g^−1^ for IR-PAN-a/SWCNT/PDPAC_ac_ and 361 F∙g^−1^ for IR-PAN-a/SWCNT/PDPAC_alk_ coatings with Coulombic efficiency ŋ = 100% (Table 2 and Table 3).

It is characteristic that the use of the flexible strips of AGF with a roughened surface as a current collector leads to a significant improvement of electrochemical characteristics of electroactive composite coatings due to good adhesion, as compared to the use of a smooth GC substrate [57,58].

Before use, the smooth surface of the original graphite foil is activated by anodic treatment. The special feature of this foil is its high porosity and low specific weight close to 1 g/cm^2^. However, when rolling the foil, the graphene nanosheets are shifted and the internal pores are closed. During anodic treatment, the surface is etched and access to internal pores is opened. It is essential that oxygen-containing functional groups are formed on graphene nanosheets during etching. When applying the composite by pouring, these groups form hydrogen bonds with composite components. The composite fills the surface pores of the current collector in the form of thin layers. In general, a strongly bonded composite coating is formed.

Figure 7 and Figure 8 show CV of AGF-based electrode materials recorded at potential scan rate of 5 mV·s^−1^, as well as galvanostatic charge-discharge dependences at the discharge current density of 0.5, 1.5, 3.0 mA·cm^−2^ in 1 M LiClO_4_ in propylene carbonate. Redox transitions on CV of AGF/IR-PAN-a/PDPAC and AGF/IR-PAN-a/SWCNT/PDPAC electrodes are noticeable when the potential scan rate is reduced to 5 mV·s^−1^ due to sufficient time for ions to access electroactive areas [62]. The CV shape for the studied coatings indicates the predominant contribution of the electric double layer charging to the specific capacitance.

The CV of the AGF/IR-PAN-a/PDPAC_alk_ electrode shows a wide anode wave at 1.0 V and a corresponding wide cathode peak in the range of 0.62 V associated with redox transitions in the polymer (Figure 7). In the AGF/IR-PAN-a/SWCNT/PDPAC_alk_, redox transitions typical of the polymer are not observed due to the predominant contribution of the double layer capacitance. CV of the IR-PAN-a/PDPAC_ac_ and IR-PAN-a/SWCNT/PDPAC_ac_ coatings on AGF demonstrate more pronounced redox transitions of the polymer due to the Faraday pseudocapacitance.

Specific surface capacitances C_s_ of AGF-based electrode materials in 1 M LiClO_4_ in propylene carbonate calculated from CV are given in Table 2. For ternary nanocomposites, as compared to binary ones, the capacitances increase from 0.135 to 0.151 F∙cm^−2^ (by 12% for IR-PAN-a/SWCNT/PDPAC_alk_) and from 0.129 to 0.161 F∙cm^−2^ (by 25% for IR-PAN-a/SWCNT/PDPAC_ac_). Coulombic charge-discharge efficiency ŋ of AGF/IR-PAN-a/PDPAC and AGF/IR-PAN-a/SWCNT/PDPAC electrodes is close to 100%.

When calculating the specific weight capacitance C_w_ of composite coatings deposited on AGF, the capacitances introduced by AGF were subtracted from the total electrode capacitance at the corresponding charge-discharge currents [24]. The capacitance loss during charge-discharge of binary IR-PAN-a/PDPAC composite coatings on AGF at 0.5 mA·cm^−2^ during the first 10 cycles is 7–9%, after the tenth cycle no drop in capacitance is observed. The AGF/IR-PAN-a/SWCNT/PDPAC_ac_ electrode works stably with the capacity loss of 1.4% in the first 10 charge-discharge cycles at 0.5 mA·cm^−2^.

In the simultaneous presence of the three components of PDPAC, IR-PAN-a and SWCNT in a nanocomposite coating, capacitance characteristics of the electrodes are 361, 239, 142 F/g (AGF/IR-PAN-a/SWCNT/PDPAC_alk_) and 394, 283, 180 F/g (AGF/IR-PAN-a/SWCNT/PDPAC_ac_) at charge-discharge currents of 0.5, 1.5, 3.0 mA·cm^−2^ (Table 3). For the IR-PAN-a/SWCNT/PDPAC_alk_ coating, the capacitance retention at the increase in charge-discharge currents to 1.5 and 3.0 mA·cm^−2^ amounts to 66 and 39%, whereas the capacitance of the AGF/IR-PAN-a/SWCNT/PDPAC_ac_ electrode is retained at 72 and 46%. Combining the polymer with CNT can reduce degradation of the polymer component caused by volume change of electroactive coating during cycling.

Characteristic features of IR-PAN-a/SWCNT/PDPAC nanocomposite coatings are high electrical conductivity (up to 7.2 × 10^−3^ S/cm) and highly developed surface. This simultaneously provides conditions for creating a double-layer capacitance and pseudocapacitance due to the rapid diffusion of electrolyte ions at the electrode–electrolyte interface. Taken together, this leads to an increase in cycling stability of the coatings and an improvement in electrochemical characteristics.

## 3. Experimental

### 3.1. Materials

Diphenylamine-2-carboxylic acid (DPAC) (C_13_H_11_O_2_N) (analytical grade), (NH_4_)_2_SO_4_ (Fisher Chemical), sulfuric acid (reagent grade), formic acid (FA) (analytical grade), aqueous ammonia (reagent grade), and chloroform (reagent grade) were used without any additional purification. Ammonium persulfate (analytical grade) was purified using recrystallization from distilled water. Propylene carbonate was dried over molecular sieves. LiClO_4_ (Aldrich) was dried in vacuum at 120 °C for 3 days. The electrolyte prepared from a 1 M LiClO_4_ solution in propylene carbonate was stored under argon.

SWCNTs from Carbon Chg, Ltd. (Moscow, Russia) with values of *d* = 1.4–1.6 nm, *l* = 0.5–1.5 µm were produced by electric arc discharge technique with Ni/Y catalyst. To prepare IR-PAN-a, the suspension of IR-heated polyacrylonitrile in the KOH aqueous solution was dried at 80 °C in a vacuum, and the powder was IR heated at 800 °C for 2 min in a nitrogen atmosphere [59]. The GC-2000 glassy carbon plates (NIIgrafit, Moscow, Russia) sized 0.5 × 3 cm was polished with a diamond paste of the ASM-3/2 type. To produce AGF, the graphite foil (GF) (Unichimtek, MSU, Russia) was used. The GF strips sized 5 × 0.5 cm were anodized in 0.1 M (NH_4_)_2_SO_4_ electrolyte for 4 min at 3.0 V and 0.3 A [63]. 

### 3.2. Synthesis of IR-PAN-a/PDPAC and IR-PAN-a/SWCNT/PDPAC

IR-PAN-a/PDPAC composites were prepared via oxidative polymerization of DPAC in the presence of 10 wt% IR-PAN-a in the homogeneous acidic medium (IR-PAN-a/PDPAC_ac_) and in the heterophase system in an alkaline medium (IR-PAN-a/PDPAC_alk_) described in [59].

IR-PAN-a/SWCNT/PDPAC nanocomposites were synthesized in two different ways according to the synthesis method for IR-PAN-a/PDPAC in [59]. The content of IR-PAN-a and SWCNT was equal (C_IR-PAN-a_ = C_SWCNT_ = 10 wt% relative to the monomer weight).

### 3.3. Electrodes Preparation

The electroactive coatings made of suspensions of nanocomposites in FA were applied to the GC and AGF substrates. An HD 3200 ultrasonic homogenizer was used to sonicate the suspensions of nanocomposites in FA (1.5 wt%). The coating square was 1 cm^2^.

### 3.4. Electrochemical Measurements

Cyclic voltammograms (CV) and galvanostatic charge-discharge curves in the potential range of −0.5–1.4 V were recorded using an IPC-Compact P-8 potentiostat (Elins, Russia). Electrochemical measurements were made in a sealed three-electrode cell in the argon atmosphere in a 1 M LiClO_4_ solution in propylene carbonate. The Pt plate (1 cm^2^) was used as an auxiliary electrode. The Ag/AgCl was used as a reference electrode.

According to the method described in [57], coulombic efficiency ŋ, specific weight and surface capacitances C_w_ and C_s_ were calculated from the charge-discharge curves.

### 3.5. Materials Characterization

Attenuated total reflection (ATR) FTIR spectra were recorded using a HYPERION-2000 IR microscope (Bruker, Karlsruhe, Germany) coupled with the Bruker IFS 66v FTIR spectrometer (Karlsruhe, Germany) in the range of 600–4000 cm^−1^ (ZnSe crystal, resolution of 2 cm^−1^).

Raman spectra were recorded using a Senterra II Raman spectrometer (Bruker, Karlsruhe, Germany). A laser with a wavelength of 532 nm and a power of 0.25 mW was used. The spectral resolution was 4 cm^−1^.

An XRD analysis was performed using a Difray-401 X-ray diffractometer (Scientific Instruments Joint Stock Company, Saint Petersburg, Russia) with Bragg–Bretano focusing on Cr*K*_α_ radiation, *λ* = 0.229 nm.

FE-SEM images were taken using a Zeiss Supra 25 FE-SEM field emission scanning electron microscope (Carl Zeiss AG, Jena, Germany).

Electric characteristics of the nanocomposites were measured using the Miller FPP-5000 4-Point Probe (Fountain Valley, CA, USA).

## 4. Conclusions

Electrochemical behavior of the advanced electrodes based on the IR-PAN-a/SWCNT/PDPAC ternary nanocomposite coatings compared to IR-PAN-a/PDPAC binary one on roughened AGF substrate in 1 M LiClO_4_ electrolyte in propylene carbonate were studied for the first time. Ternary nanocomposites for electrode coatings were prepared via in situ oxidative polymerization of DPAC in the presence both of IR-PAN-a and SWCNT in two different ways in acidic and alkaline media. The main contribution to the electrochemical capacitance is provided by the electric double-layer charging and specific weight capacitances C_w_ of electrodes reach 394, 283, 180 F∙g^−1^ (AGF/IR-PAN-a/SWCNT/PDPAC_ac_) and 361, 239, 142 F∙g^−1^ (AGF/IR-PAN-a/SWCNT/PDPAC_alk_) at charge-discharge currents of 0.5, 1.5, 3.0 mA∙cm^−2^. Specific surface capacitances C_s_ of hybrid electrodes amount to 0.161 and 0.151 F∙cm^−2^ for AGF/IR-PAN-a/SWCNT/PDPAC_ac_ and AGF/IR-PAN-a/SWCNT/PDPAC_alk_. Such electroactive nanocomposite coatings for hybrid electrodes are promising as a cathode material for SCs with increased voltage.

## Figures and Tables

**Figure 1 molecules-28-05093-f001:**
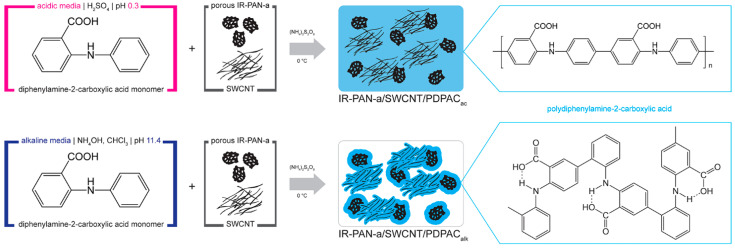
The synthesis scheme of IR-PAN-a/SWCNT/PDPAC nanocomposites.

**Figure 2 molecules-28-05093-f002:**
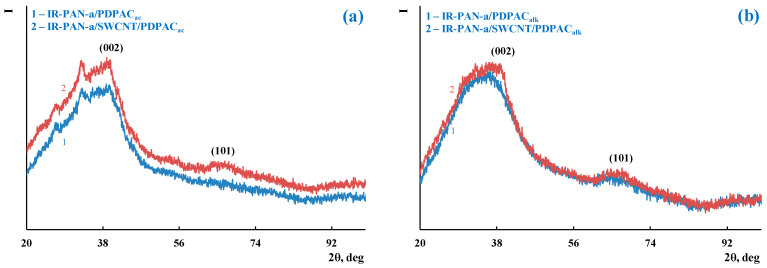
XRD of IR-PAN-a/PDPAC (1) and IR-PAN-a/SWCNT/PDPAC (2), prepared in an acidic (**a**) and alkaline media (**b**).

**Figure 3 molecules-28-05093-f003:**
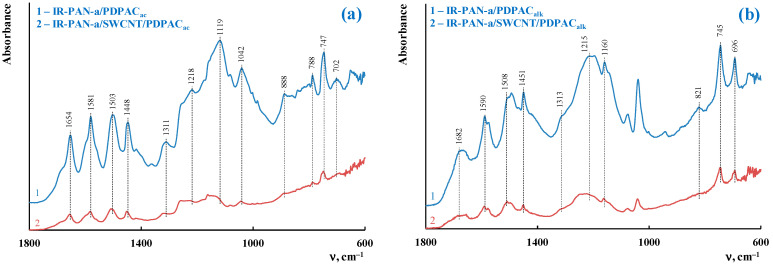
Attenuated total reflection (ATR) FTIR spectra of IR-PAN-a/PDPAC (1) and IR-PAN-a/SWCNT/PDPAC (2), prepared in an acidic (**a**) and alkaline media (**b**).

**Figure 4 molecules-28-05093-f004:**
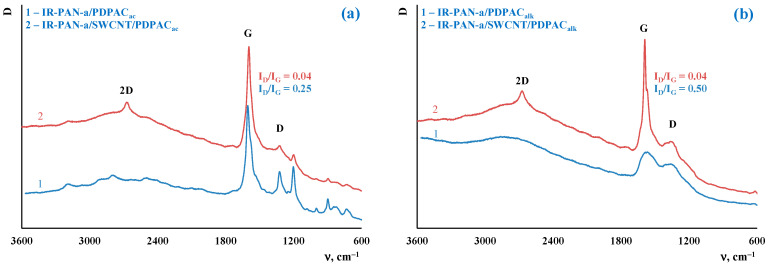
Raman spectra of IR-PAN-a/PDPAC (1) and IR-PAN-a/SWCNT/PDPAC (2), prepared in an acidic (**a**) and alkaline media (**b**).

**Figure 5 molecules-28-05093-f005:**
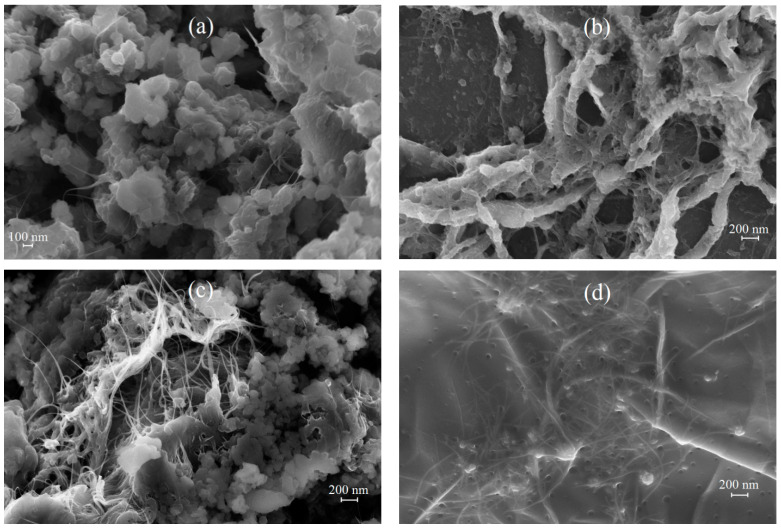
FE-SEM images of IR-PAN-a/SWCNT/PDPAC_ac_ (**a**), AGF/IR-PAN-a/SWCNT/PDPAC_ac_ (**b**), IR-PAN-a/SWCNT/PDPAC_alk_ (**c**) and AGF/IR-PAN-a/SWCNT/PDPAC_alk_ (**d**).

**Figure 6 molecules-28-05093-f006:**
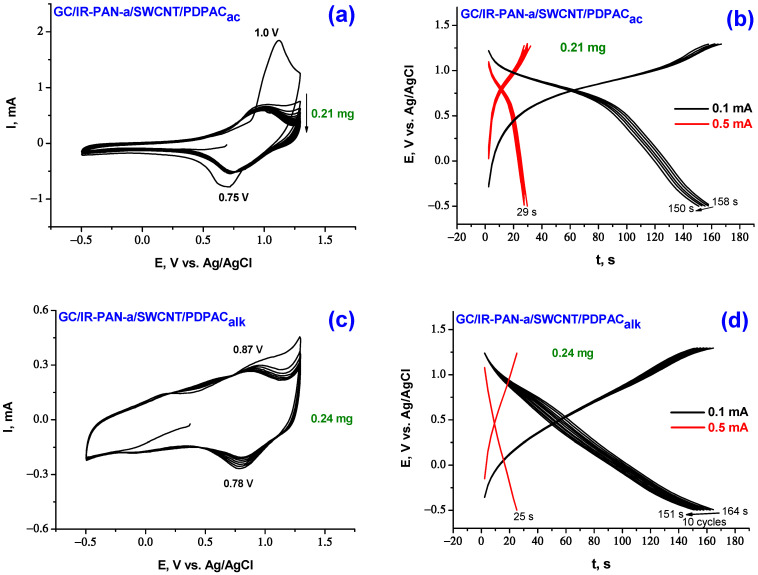
CV curves (**a**,**c**) and galvanostatic charge-discharge dependences (**b**,**d**) of the electrodes of GC/IR-PAN-a/SWCNT/PDPAC_ac_ (**a**,**b**) and GC/IR-PAN-a/SWCNT/PDPAC_alk_ (**c**,**d**) at 20 mV·s^−1^.

**Figure 7 molecules-28-05093-f007:**
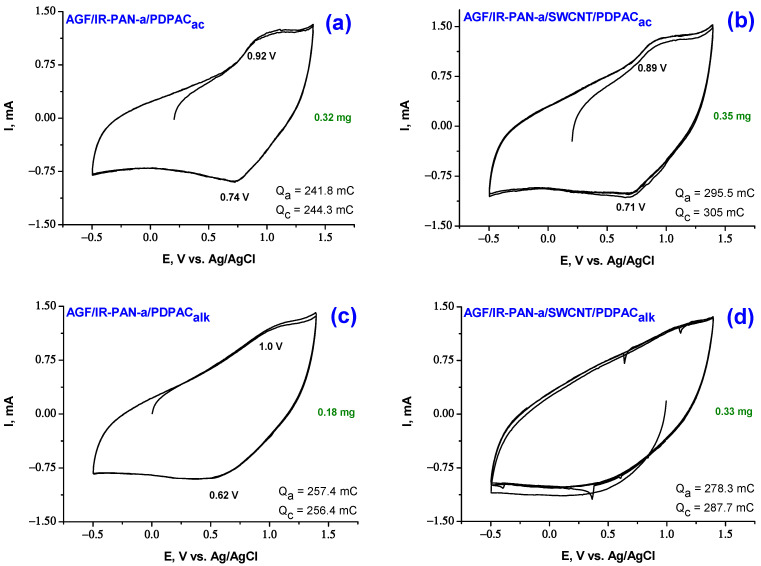
CV curves on the electrodes of AGF/IR-PAN-a/PDPAC_ac_ (**a**), AGF/IR-PAN-a/SWCNT/PDPAC_ac_ (**b**), AGF/IR-PAN-a/PDPAC_alk_ (**c**) and AGF/IR-PAN-a/SWCNT/PDPAC_alk_ (**d**) at 5 mV·s^−1^.

**Figure 8 molecules-28-05093-f008:**
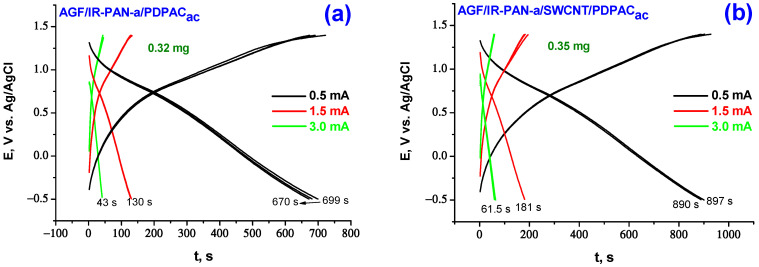
Galvanostatic charge-discharge dependences of the electrodes of AGF/IR-PAN-a/PDPAC_ac_ (**a**), AGF/IR-PAN-a/SWCNT/PDPAC_ac_ (**b**), AGF/IR-PAN-a/PDPAC_alk_ (**c**) and AGF/IR-PAN-a/SWCNT/PDPAC_alk_ (**d**) at 0.5, 1.5, 3.0 mA∙cm^−2^.

**Table 1 molecules-28-05093-t001:** The conductivity values of materials.

Materials	C_IR-PAN-a_, wt%	C_SWCNT_, wt%	σ, S/cm
IR-PAN-a/PDPAC_ac_	10	–	1.3 × 10^−5^
* SWCNT/PDPAC_ac_	–	10	2.5 × 10^−3^
IR-PAN-a/SWCNT/PDPAC_ac_	10	10	7.2 × 10^−3^
IR-PAN-a/PDPAC_alk_	10	–	1.5 × 10^−10^
* SWCNT/PDPAC_alk_	–	10	2.9 × 10^−4^
IR-PAN-a/SWCNT/PDPAC_alk_	10	10	4.8 × 10^−3^

* From [57].

**Table 2 molecules-28-05093-t002:** Electrochemical characteristics of electrode materials in 1 M LiClO_4_ in propylene carbonate, calculated from CV.

Electrode Materials	Quantity of Electricity *Q*, mC		Coulombic Efficiency ŋ, %	Specific Surface Capacitance C_s_, F∙cm^−2^
	*Q* _charge_	*Q* _discharge_		
GC/IR-PAN-a/SWCNT/PDPAC_ac_	16.8	16.5	98	9.2 × 10^−3^
GC/IR-PAN-a/SWCNT/PDPAC_alk_	17.1	15.9	93	8.8 × 10^−3^
AGF/IR-PAN-a/PDPAC_ac_	241.8	244.3	100	0.129
* AGF/SWCNT/PDPAC_ac_	269.6	270.8	100	0.145
AGF/IR-PAN-a/SWCNT/PDPAC_ac_	295.5	305.0	100	0.161
AGF/IR-PAN-a/PDPAC_alk_	257.4	256.4	99.6	0.135
* AGF/SWCNT/PDPAC_alk_	273.5	262.2	96	0.138
AGF/IR-PAN-a/SWCNT/PDPAC_alk_	278.3	287.7	100	0.151

ν = 5 mV·s^−1^ (for AGF-based) and 20 mV·s^−1^ (for GC-based). * From [57].

**Table 3 molecules-28-05093-t003:** Electrochemical characteristics of electrode materials in 1 M LiClO_4_ in propylene carbonate, calculated from charge-discharge curves.

Electrode Materials	Coatings Weight, mg	Discharge Current Density I_charge-discharge_, mA∙cm^−2^	Specific Surface Capacitance C_s_, F∙cm^−2^	Specific Weight Capacitance * C_w_, F∙g^−1^
GC/IR-PAN-a/SWCNT/PDPAC_ac_	0.21	0.1	9.2 × 10^−3^	40
		0.5		38
GC/IR-PAN-a/SWCNT/PDPAC_alk_	0.24	0.1	8.8 × 10^−3^	35
		0.5		29
AGF/IR-PAN-a/PDPAC_ac_	0.32	0.5	0.176	247
		1.5	0.103	178
		3.0	0.068	122
** AGF/SWCNT/PDPAC_ac_	0.32	0.5	0.237	438
		1.5	0.158	350
		3.0	0.112	259
AGF/IR-PAN-a/SWCNT/PDPAC_ac_	0.35	0.5	0.235	394
		1.5	0.145	283
		3.0	0.092	180
AGF/IR-PAN-a/PDPAC_alk_	0.18	0.5	0.180	461
		1.5	0.112	367
		3.0	0.076	261
** AGF/SWCNT/PDPAC_alk_	0.24	0.5	0.185	367
		1.5	0.090	183
		3.0	0.050	88
AGF/IR-PAN-a/SWCNT/PDPAC_alk_	0.33	0.5	0.216	361
		1.5	0.125	239
		3.0	0.076	142

* C_w_ calculated from coating weight. ** From [57].

## Data Availability

Not applicable.

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
