# Peer review of "Novel Hybrid Electrode Coatings Based on Conjugated Polyacid Ternary Nanocomposites for Supercapacitor Applications"

_molecules, 2023, doi:10.3390/molecules28135093_

Round 1

Reviewer 1 Report

In this manuscript, the authors reported oxidative polymerization  of diphenylamine-2-carboxylic acid (DPAC) in the presence of activated IR-pyrolyzed polyacrylo-  nitrile (IR-PAN-a) Specific weight capacitances Cw of electrodes with ternary coatings reach 394, 283, 180 F∙g –1 (AGF/IR-PAN-a/SWCNT/PDPACac) and 361, 239, 142 F∙g–1 (AGF/IR-PAN-a/SWCNT/PDPACalk) at 0.5, 1.5, 3.0 mА∙cm–2 in an aprotic electrolyte. It is well-written manuscript however there is a lack of critical discussions.

 Therefore, I recommend a minor revision of the current manuscript for publication in the Molecules. The following points should be addressed: 

1.    The graphical abstract should be presented by the scheme. It will improve the quality of the manuscript.

2.    The material synthesis process should be presented by the scheme. It will improve the quality of the manuscript.

3.    The introduction part should be extended to upgrade the importance of this work and the novelty of the paper should be clearly stated.

4.    The preparation advantages of processed materials over the previous reports should be provided.

5.    The G/D ratio should be measured and included in the manuscript

6.     The thickness of working electrodes should be included in the manuscript?

7.    First of all, as-prepared materials show a battery-type storage mechanism rather than direct capacitance behavior. That is the reason the author provides high energy and power density. Based on the GCD curves of the GC/IR-PAN-a/SWCNT/PDPACac electrodes, I see the material shows a battery-type behavior. This is likely to have the battery-type feature of the as-prepared electrodes. Please consider these comments, re-calculate the capacitance, energy and power densities and cite these critical papers. 1) Simon, Patrice, Yury Gogotsi, and Bruce Dunn. "Where do batteries end and supercapacitors begin?." Science 343.6176 (2014): 1210-1211. 2) Simon, Patrice, and Yury Gogotsi. "Perspectives for electrochemical capacitors and related devices." Nature Materials 19.11 (2020): 1151-1163. 3) https://doi.org/10.1016/j.jallcom.2023.170489 4) https://doi.org/10.1016/j.ijhydene.2023.04.159 5) https://doi.org/10.1021/acs.energyfuels.2c04273

 8.    The writing style, grammar and language usage should be checked by a native speaker.

The writing style, grammar and language usage should be checked by a native speaker.

Author Response

The authors are grateful to the reviewer for constructive and valuable comments on the manuscript. Please find below our answers to the comments.

Comments and Suggestions for Authors

  1. The graphical abstract should be presented by the scheme. It will improve the quality of the manuscript.

The graphical abstract was presented by the scheme.

  1. The material synthesis process should be presented by the scheme. It will improve the quality of the manuscript.

The scheme of material synthesis process was presented.

Figure 1. The synthesis scheme of IR-PAN-a/SWCNT/PDPAC nanocomposites. .

Appropriate additions were introduced into the text in colored characters.

  1. The introduction part should be extended to upgrade the importance of this work and the novelty of the paper should be clearly stated.

The introduction part was extended.

  1. The preparation advantages of processed materials over the previous reports should be provided.

In the last decade, researchers have focused on the production of hybrid electrode materials that include, along with a conductive polymer, a combination of different carbon nanomaterials. Only a few papers have shown that PANI-based nanocomposites containing graphene and CNT nanoparticles have improved SCs.

Attention should be drawn to the fact that the information available in the literature on the study of electrode materials based on ternary nanocomposites of PANI with two carbon components describes the results of electrochemical measurements, conducted in acid or alkaline aqueous electrolytes. We have not been able to find any references to studies of such nanocomposites in lithium organic electrolytes. The main disadvantages of aqueous electrolytes are low discharge voltage, narrow operating temperature range, high corrosive activity. On the other hand, lithium batteries use organic electrolytes, which have a wider range of operating potentials and operating temperatures, high corrosion resistance. It should be noted that the number of works on SCs with organic electrolytes is extremely limited (some of them were done by the co-authors of this article [60. React. Funct. Polym. 2022, 173, 105225. 10.1016/j.reactfunctpolym.2022.105225, 61. Polymers 2023, 15, 1896. https://doi.org/10.3390/polym15081896]). Moreover, we have not been able to find any research work on electrode materials based on ECP ternary nanocomposites with two carbon components in lithium organic electrolytes. Nevertheless, the transition to organic electrolytes gives the prospect of creating hybrid devices that combine the advantages of SCs and lithium batteries.

The present work is the first study of the electrochemical behavior of a cathode material based on a conjugated polyacid, IR-PAN-a and CNT in a lithium organic electrolyte. Such cathode materials in organic electrolytes are the most promising for the creation of hybrid SCs due to the possibility of increasing the SC voltage and achieving high values of energy density and charge-discharge current.

The choice of polydiphenylamine-2-carboxylic acid (PDPAC) as a polymer component is due to the possibility of coordination of carbon nanoparticles not only for amine but also for carboxylic groups. Furthermore, the presence of carboxylic groups makes it difficult to aggregate polymer chains (as in the case of PANI). Steric difficulties caused by carboxylic groups contribute to a looser structure that facilitates electrolyte penetration.

The use of the flexible strips of anodized graphite foil (AGF) with a roughened surface as a current collector leads to a significant improvement of electrochemical characteristics of electroactive composite coatings due to good adhesion, as compared to the use of a smooth glass carbon substrate.

Appropriate additions were introduced into the text in colored characters.

  1. The G/D ratio should be measured and included in the manuscript

Raman spectra of composites were recorded to measure the G/D ratio.

Appropriate additions were introduced into the text in colored characters.

  1. The thickness of working electrodes should be included in the manuscript?

An important feature of this work is the use of the flexible strip of anodized graphite foil (AGF) with a roughened surface as a current collector. Before use, the smooth surface of the original graphite foil was activated by anodic treatment. The special feature of this foil is its high porosity and low specific weight close to 1 g/cm2. However, when rolling the foil, the graphene nanosheets are shifted and the internal pores are closed. During anodic treatment, the surface is etched and access to internal pores is opened. It is essential that oxygen-containing functional groups are formed on graphene nanosheets during etching. When applying the composite by pouring, these groups form hydrogen bonds with composite components. The composite fills the surface pores of the current collector in the form of thin layers. In general, a strongly bonded composite coating is formed. Since the depth of penetration of the composite into the current collector is unknown, the graphite foil thickness is chosen for practical reasons. In our case, it was 0.8 mm. The coating weight was determined by weighing.

Appropriate additions were introduced into the text in colored characters.

  1. First of all, as-prepared materials show a battery-type storage mechanism rather than direct capacitance behavior. That is the reason the author provides high energy and power density. Based on the GCD curves of the GC/IR-PAN-a/SWCNT/PDPACac electrodes, I see the material shows a battery-type behavior. This is likely to have the battery-type feature of the as-prepared electrodes. Please consider these comments, re-calculate the capacitance, energy and power densities and cite these critical papers. 1) Simon, Patrice, Yury Gogotsi, and Bruce Dunn. "Where do batteries end and supercapacitors begin?." Science 343.6176 (2014): 1210-1211. 2) Simon, Patrice, and Yury Gogotsi. "Perspectives for electrochemical capacitors and related devices." Nature Materials 19.11 (2020): 1151-1163. 3) https://doi.org/10.1016/j.jallcom.2023.170489 4) https://doi.org/10.1016/j.ijhydene.2023.04.159 5) https://doi.org/10.1021/acs.energyfuels.2c04273

Indeed, on the discharge curves recorded on the GC current collector, extended sections with a slight slope (plateau) are observed, when the electrode potential changes slightly during discharge. These coatings are highly unstable and degraded during cycling. They have been investigated to determine the nature of surface redox reactions in the composite. The electrochemical capacity of such coatings is very low. Therefore, coatings at AGF have been further investigated. In this case, CV of the coatings acquired a quasi-rectangular shape, typical of pseudocapacitors with a predominance of the contribution of double-layer charging. Redox transitions on CV are weakly expressed. In the Fig. 7, GCD dependencies are almost straightforward and thus correspond to the behavior of the pseudocapacitor. We emphasize that the contribution of pseudocapacitance can be detected during charge-discharge with low current densities. But such modes are of no practical interest for pseudocapacitors. Therefore, they have not been studied.

Proposed references were cited.

  1. The writing style, grammar and language usage should be checked by a native speaker.

A professional translator has corrected typos and mistakes.

Reviewer 2 Report

The manuscript is well organized but the long title. Therefore, I suggest the authors to simplify it before publication.

This paper by Sveta Ozkan and others described the Supercapacitor Applications using Conjugated Polyacid Nanocomposites with Activated IR-Pyrolyzed Polyacrylonitrile and Single-Walled Carbon Nanotubes. Both of these materials and such concept has already been reported by other groups (Reactive and Functional Polymers 173 (2022): 105225, and several others). Please cite them and stated the creative point of this work. Furthermore, the following questions need to be well elucidated.
1. The XRD needs to be indexed.
2. The CVs shape of the electrodes here in Figure 5 and Figure 6 are a bit different from a Faraday pseudocapacitance CV shape. I would like to see the CV shape at low scan rate. Can authors please translate the y-axis into F/g? so that the CVs at low scan rates can be more readable.
3. There are a lot of electroactive materials coating SWCNTs as electrode materials, such as Reactive and Functional Polymers 173 (2022): 105225, having been investigated. Why choose PDPAC, and IR-PAN-a coating SWCNTs?
4. The authors have to compare their results with previous literature data.
5. The discussion on the CV analysis is quite limited and reasons for the changing of peak locations and shapes are listed as prepare electrode in acidic medium and alkaline medium. This is not an explanation for the behavior and needs to be improved.

Author Response

The authors are grateful to the reviewer for constructive and valuable comments on the manuscript. Please find below our answers to the comments.

Comments and Suggestions for Authors

The manuscript is well organized but the long title. Therefore, I suggest the authors to simplify it before publication.

Appropriate changes were made.

The new title was suggested.

Novel Hybrid Electrode Coatings Based on Conjugated Polyacid Ternary Nanocomposites for Supercapacitor Applications  

This paper by Sveta Ozkan and others described the Supercapacitor Applications using Conjugated Polyacid Nanocomposites with Activated IR-Pyrolyzed Polyacrylonitrile and Single-Walled Carbon Nanotubes. Both of these materials and such concept has already been reported by other groups (Reactive and Functional Polymers 173 (2022): 105225, and several others). Please cite them and stated the creative point of this work.

In the last decade, researchers have focused on the production of hybrid electrode materials that include, along with a conductive polymer, a combination of different carbon nanomaterials. Only a few papers have shown that PANI-based nanocomposites containing graphene and CNT nanoparticles have improved SCs.

Attention should be drawn to the fact that the information available in the literature on the study of electrode materials based on ternary nanocomposites of PANI with two carbon components describes the results of electrochemical measurements, conducted in acid or alkaline aqueous electrolytes.

We have not been able to find any references to studies of such nanocomposites in lithium organic electrolytes. The main disadvantages of aqueous electrolytes are low discharge voltage, narrow operating temperature range, high corrosive activity. On the other hand, lithium batteries use organic electrolytes, which have a wider range of operating potentials and operating temperatures, high corrosion resistance. It should be noted that the number of works on SCs with organic electrolytes is extremely limited (some of them were done by the co-authors of this article [60. React. Funct. Polym. 2022, 173, 105225. 10.1016/j.reactfunctpolym.2022.105225, 61. Polymers 2023, 15, 1896. https://doi.org/10.3390/polym15081896]). Moreover, we have not been able to find any research work on electrode materials based on ECP ternary nanocomposites with two carbon components in lithium organic electrolytes. Nevertheless, the transition to organic electrolytes gives the prospect of creating hybrid devices that combine the advantages of SCs and lithium batteries.

The present work is the first study of the electrochemical behavior of a cathode material based on a conjugated polyacid, IR-PAN-a and CNT in a lithium organic electrolyte. Such cathode materials in organic electrolytes are the most promising for the creation of hybrid SCs due to the possibility of increasing the SC voltage and achieving high values of energy density and charge-discharge current.

The choice of polydiphenylamine-2-carboxylic acid (PDPAC) as a polymer component is due to the possibility of coordination of carbon nanoparticles not only for amine but also for carboxylic groups. Furthermore, the presence of carboxylic groups makes it difficult to aggregate polymer chains (as in the case of PANI). Steric difficulties caused by carboxylic groups contribute to a looser structure that facilitates electrolyte penetration.

The use of the flexible strips of anodized graphite foil (AGF) with a roughened surface as a current collector leads to a significant improvement of electrochemical characteristics of electroactive composite coatings due to good adhesion, as compared to the use of a smooth glass carbon substrate.

Appropriate additions were introduced into the text in colored characters.

  1. The XRD needs to be indexed.

Appropriate additions were introduced into the XRD figures.

Diffractograms of binary and ternary composites identify reflection peaks of IR-PAN-a in the range of scattering angles 2θ = 39°, 69° (CrKα radiation). These diffraction peaks correlate to Miller indices (002), (101). The carbon phase reflection peak from a single SWCNT plane is not identified.

  1. The CVs shape of the electrodes here in Figure 5 and Figure 6 are a bit different from a Faraday pseudocapacitance CV shape. I would like to see the CV shape at low scan rate. Can authors please translate the y-axis into F/g? so that the CVs at low scan rates can be more readable.

We have not investigated CV and galvanostatic charge-discharge curves for GC coatings at low potential scan rates and low charge-discharge currents. Coatings on GC have poor adhesion and degrade quickly. Our goal was to detect redox transitions in composites and compare them with literature data on the nature of these redox transitions. Later, the capacitive characteristics of composite coatings were studied on activated graphite foil. In the CV of coatings, the peaks of the redox transitions were smoothed out, and the quasi-rectangular shape of the CV indicated the predominant contribution of the double layer charging capacitance. The discharge curves at low charge-discharge currents retained signs of Faraday reactions. In general, this picture is due to the synchronization of Faraday reactions and double-layer charging.

  1. There are a lot of electroactive materials coating SWCNTs as electrode materials, such as Reactive and Functional Polymers 173 (2022): 105225, having been investigated. Why choose PDPAC, and IR-PAN-a coating SWCNTs?

Indeed, there is a lot of information in the literature about the electrochemical behavior of electrode coatings that contain CNTs. However, most of the papers published the results of electrochemical studies carried out in proton electrolytes. There have been very few studies of electrochemical properties in organic electrolytes. Some of them were done by the authors of this article. The article referred to by the reviewer also belongs to the authors [560].

The choice of polydiphenylamine-2-carboxylic acid (PDPAC) as a polymer component is due to the possibility of coordination of carbon nanoparticles not only for amine but also for carboxylic groups. Furthermore, the presence of carboxylic groups makes it difficult to aggregate polymer chains (as in the case of PANI). Steric difficulties caused by carboxylic groups contribute to a looser structure that facilitates electrolyte penetration.

As for the choice of "IR-PAN-a coating SWCNTs", when producing ternary nanocomposites, we performed oxidative polymerization of DPAC in the presence of IR-PAN-a and SWCNT simultaneously. We did not prepare the IR-PAN-a/SWCNT composite beforehand. As the second carbon component we used IR-PAN, leached to form a highly porous structure, adsorbing a part of the polymer phase, resulting in the general loosening of the nanocomposite.

Appropriate additions were introduced into the text in colored characters.

  1. The authors have to compare their results with previous literature data.

It would be correct to compare the obtained results with the previously published ones if the ternary nano-composites were obtained under the same conditions. Unfortunately, we have not been able to find any work in which porous carbon was used as one of the carbon components, as well as papers in which the results of a study of electrochemical characteristics of composites in aprotonic organic electrolytes were presented.

  1. The discussion on the CV analysis is quite limited and reasons for the changing of peak locations and shapes are listed as prepare electrode in acidic medium and alkaline medium. This is not an explanation for the behavior and needs to be improved.

Appropriate additions were introduced into the text in colored characters.

Reviewer 3 Report

Comments

The paper has been written very well and presents some good interesting studies. The paper does require minor changes or modifications.

I will suggest accepting this paper after minor revision. This manuscript successfully prepares Capacitive Characteristics of Novel Hybrid Electrode Coatings Based on Conjugated Polyacid Nanocomposites with Activated IR-Pyrolyzed Polyacrylonitrile and Single-Walled Carbon Nanotubes for Supercapacitor Applications.

The following points need to be addressed before publication:

1.     The introduction should be moderately modified. I suggest the authors remove some unnecessary old citations and be up to date in his/her research and add some necessary statements, and elaborate in more scientific coherence.

2.     There are some insufficient typos and mistakes in the text. Please check this carefully.

3.     Recently published references in the field of supercapacitors should be cited in the introduction part.

https://doi.org/10.1021/acsaem.0c00874

https://doi.org/10.1016/B978-0-12-817445-6.00010-7

Author Response

The authors are grateful to the reviewer for constructive and valuable comments on the manuscript. Please find below our answers to the comments.

Comments and Suggestions for Authors

  1. The introduction should be moderately modified. I suggest the authors remove some unnecessary old citations and be up to date in his/her research and add some necessary statements, and elaborate in more scientific coherence.

The introduction part was modified. Some old citations have been replaced with new ones.

  1. There are some insufficient typos and mistakes in the text. Please check this carefully.

A professional translator has corrected typos and mistakes.  

  1. Recently published references in the field of supercapacitors should be cited in the introduction part.

https://doi.org/10.1021/acsaem.0c00874

https://doi.org/10.1016/B978-0-12-817445-6.00010-7

Proposed references were cited,
